# Nature’s Preservative: Epigallocatechin Gallate-Loaded Edible Film Extends Mango Shelf Life

**DOI:** 10.3390/foods14213609

**Published:** 2025-10-23

**Authors:** Gerui Ren, Lei Liu, Miaomiao Wang, Junjie Pan, Zhoutao Wang, Ruiqi Hu, Junmei Zhou, Xin Song, Kejun Cheng, Wenliang Cheng

**Affiliations:** 1Chemical Biology Center, Lishui Institute of Agriculture and Forestry Sciences, Lishui 323000, China; rengerui@zjgsu.edu.cn (G.R.);; 2School of Food Science and Biotechnology, Zhejiang Gongshang University, Hangzhou 310018, China; 3School of Pharmaceutical Sciences, Fuchun Campus, Zhejiang Chinese Medical University, Hangzhou 311402, China

**Keywords:** EGCG, chitosan, collagen hydrolysate, mango preservation, edible coating

## Abstract

To reduce the environmental impact of plastic packaging in the fruit supply chain, this study developed an edible natural CH-CS-EGCG coating (collagen hydrolysate-chitosan-epigallocatechin gallate) for mango preservation. The coating was prepared using an optimized CH:CS mass ratio of 1:4 with 3 wt% EGCG, exhibiting enhanced mechanical properties and low water vapor permeability. SEM and FT-IR analyses confirmed the successful incorporation of EGCG into the CH-CS matrix through hydrogen bonding, hydrophobic interactions, and electrostatic forces. The CH-CS-EGCG coating demonstrated superior antioxidant activity: its ABTS radical scavenging capacity and DPPH scavenging capacity were 234% and 422% higher, respectively, than those of the CH-CS coating. It also effectively inhibited bacterial growth, achieving a 92% inhibition rate against *Staphylococcus aureus* after 24 h of incubation. When applied to mango preservation, the CH-CS-EGCG coating significantly slowed down fruit decay and deterioration, extending the shelf life by 6 days. The CH-CS-EGCG coating offers a promising eco-friendly alternative for fruit preservation, broadening the applications of EGCG and advancing research in edible coatings.

## 1. Introduction

Globally, mango (*Mangifera indica* L.) ranks as the second most economically important tropical fruit [1,2]. As a typical climacteric fruit, it is highly susceptible to spoilage during transportation and storage. In tropical and subtropical regions, post-harvest loss rates can reach as high as 50%, depending on mango varieties, post-harvest handling methods and technical levels [3,4,5]. In China alone, millions of tons of mangoes are lost annually due to improper preservation, resulting in economic losses exceeding tens of millions of dollars [6]. This issue not only impacts local and regional market sales but also hinders international trade. Therefore, adopting effective post-harvest preservation methods for mangoes is crucial.

Traditional preservation methods, such as freezing and chemical treatments, have inherent limitations. While low-temperature environments reduce microbial activity and slow fruit respiration, they can cause chilling injury [7]—an issue that compromises fruit quality. Chemical preservatives, on the other hand, raise significant concerns about chemical residues. In recent years, safety and sustainability concerns have spurred interest in biopolymer-based edible films. Formed by applying edible materials to fruit surfaces, these films act as essential physical barriers and enable physiological regulation, thereby enhancing preservation efficacy [8,9]. Polysaccharide biomacromolecules are widely used as the matrix for such films [10,11,12]. These polysaccharide molecules interact with other components of the film matrix via hydrogen bonding, hydrophobic interactions, and electrostatic interactions. These interactions enhance the polymer film’s network integrity and functionality.

Chitosan (CS), a natural polycationic polysaccharide, exhibits excellent biocompatibility, film-forming property, and functional properties such as antioxidant and antibacterial activity, leading to its widespread use in fruit preservation [13,14]. However, the mechanical properties and stability of pure chitosan films are often inadequate, further compounded by high hydrophilicity and water vapor permeability which hinder large-scale application. Consequently, researchers frequently enhance chitosan films by incorporating bioactive substances such as peptides, proteins, and polyphenols to improve quality and preservation efficacy [15,16].

Collagen hydrolysate (CH) is particularly well-suited as a modifying material due to its advantageous properties: high water solubility, low viscosity, and good polymer compatibility [17]. The non-covalent interaction between CH and CS, especially their electrostatic binding in acidic environments, can significantly improve the water vapor permeability and adhesion of chitosan films [18,19,20]. Moreover, epigallocatechin gallate (EGCG), a tea polyphenol valued for its potent antioxidant and antibacterial activity, is incorporated to further improve the functionality of chitosan films [21,22]. Dai et al. [23] found that the multiple phenolic hydroxyl groups of EGCG could form strong hydrogen bonds with the hydroxyl and amino groups on chitosan chains. Meanwhile, the hydrophobic interactions between the benzene rings of EGCG and the hydrophobic acetyl groups of chitosan could also enhance the cross-linking of chitosan. This dual interaction strengthened the network structure of the chitosan film, thereby improving its mechanical properties (notably manifested as a significant increase in tensile strength). This is particularly crucial for the development of chitosan-based edible coating materials.

This research developed a bioactive edible coating film based on a collagen hydrolysate-chitosan matrix incorporated with EGCG. The film was successfully applied to mangoes to extend their shelf life, evaluating the impact of varying EGCG concentrations on fruit quality and physiological changes throughout storage. This research provides valuable reference for the study of edible coating films in food preservation.

## 2. Materials and Methods

### 2.1. Materials

Materials were sourced as follows: fresh mango (Yonghui Supermarket, Hangzhou, China); EGCG (≥95% purity, Source Leaf Organism, Shanghai, China); collagen hydrolysate (CH, 92% purity, fish scales/skin origin, Yangsen Biotechnology, Changzhou, Jiangsu, China); chitosan (CS, 87% deacetylation), 1,1-diphenyl-2-picryl-hydrazyl radical (DPPH), and 2,2′-Azino-bis (3-ethylbenzothiazoline-6-sulfonic acid) (ABTS) (Sinopharm Chemical Reagent, Shanghai, China). All other chemicals were obtained from Macleans Co. Ltd., (Shanghai, China), unless otherwise stated.

### 2.2. Preparation of CH-CS Films

The CS (2 wt%) solution was prepared in 1% (*v*/*v*) acetic acid and degassed ultrasonically. Simultaneously, the required amount of 4 wt% CH was dissolved in distilled water at 40 °C (30 min stirring, constant-temperature bath). The CH and CS solutions were combined at room temperature at various weight ratios (from 1:1 to 1:5). Glycerol (20 wt% relative to total dry biopolymer mass) was added as a cross-linker, and the mixture was homogenized at 200 rpm for 30 min. After adjusting the pH to 3.0 with HCl and defoaming, 40 mL aliquots were cast into plastic dishes. Films were formed by drying at 30 °C for 24 h, peeled, and conditioned at 55% relative humidity (sealed oven, 48 h).

### 2.3. Preparation of CH-CS-EGCG Films

The prepared collagen hydrolysate solution and chitosan solution were added to a beaker at a mass ratio of 1:4. Then, 20 wt% glycerol was added. After stirring for 60 min at 300 r/min on a magnetic stirrer, the solution was divided into three beakers. Different amounts of EGCG (1 wt%, 2 wt%, and 3 wt%, calculated based on the total dry weight of CH and CS) were added to each beaker, respectively. At room temperature, the three solutions were stirred on a magnetic stirrer (DF-101S, Hangzhou Educational Equipment Co. Ltd., Hangzhou, China) until completely mixed. Film formation followed the previously described method, resulting in EGCG-loaded collagen hydrolysate-chitosan composite films.

### 2.4. Determination of Physical Properties of the Films

#### 2.4.1. Thickness

Film thickness was measured using a digital micrometer (Mitutoyo Co., Kawasaki, Japan). To assess spatial homogeneity, thickness values were recorded at 10 distinct locations per sample (e.g., center, near-edge regions). The arithmetic mean of these measurements represents the reported film thickness.

#### 2.4.2. Light Transmittance

The films were cut into 2 cm × 2 cm square specimens for UV-Vis analysis. The UV-2600 spectrophotometer (Shimadzu Scientific Instruments, Kyoto, Japan) was configured with a wavelength of 200 nm and a scan rate of 100 nm/min. The absorbance of each film specimen was recorded, with three replicate measurements performed for each sample. The transmittance was subsequently calculated using the Beer-Lambert law.
(1)$$\begin{array}{c} A=\mathrm{Lg}\frac{1}{T}\; \end{array}$$ where *T* is the transmittance (%) and *A* is the absorbance (%).

#### 2.4.3. Water Vapor Permeability (WVP)

The WVP values of the films were evaluated following the methodology outlined by Hua et al. [24]. A circular sample with a 6 cm diameter was securely mounted onto the lid of a 15 mL round-bottom flask containing 10 mL of distilled water. The flask was saturated with water vapor and placed in a desiccator filled with anhydrous silica gel to maintain a humidity gradient between the interior and exterior of the flask. The mass of the flask was recorded at regular intervals (every 24 h) until a steady weight was attained. The WVP values were then calculated using Equation (2) based on the collected data.
(2)$$\begin{array}{c} \mathrm{WVP}=\frac{m\times L}{A\times t\times ∆P}\; \end{array}$$ where *m* represents the mass of water that penetrated the film (g); *L* is the film thickness (m); *A* is the exposed film area (m^2^); *t* is the duration of water vapor permeation (s); Δ*P* is the water vapor pressure differential across the film (Pa).

#### 2.4.4. Mechanical Properties

The mechanical properties of the films (3 cm × 7 cm) were assessed in tension mode using a texture analyzer (TA.XT2i, Stable Micro Systems, London, UK) equipped with an A/TG probe, following the approach described by Khoshkalampour et al. [25]. A single compression process was incorporated into the tensile testing procedure, minimum trigger force was 5 g, pre-test speed was 1 mm/s, test speed was 1 mm/s, and post-test speed was 3 mm/s. The strength and ductility of the composite films were characterized by calculating their tensile strength (TS) and elongation at break (EB) using Equations (3) and (4), respectively, based on the experimental data.
(3)$$\begin{array}{c} \mathrm{TS}=\frac{F}{S}\; \end{array}$$ where TS represents the tensile strength (MPa), calculated as the maximum tensile force (*F*, *N*) sustained by the composite film at break divided by its cross-sectional area (S, mm^2^).
(4)$$\begin{array}{c} \mathrm{EB}=\frac{\left(L-L_{0}\right)}{L_{0}}\times 100\%\; \end{array}$$ where EB denotes the elongation at break (%), determined by the ratio of the final length of the film at break (*L*, mm) to its initial length (*L*_0_, mm).

### 2.5. Scanning Electron Microscope (SEM)

The microstructure of the CH-CS and CH-CS-EGCG films was examined using a Gemini SEM 300 microscope (Carl Zeiss, Oberkochen, Germany). Before imaging, the films were prepared by cutting them into 5 mm × 10 mm rectangles and mounted on an aluminum stub using conductive adhesive.

### 2.6. Fourier Transform Infrared Spectroscopy (FT-IR)

The spectra of EGCG, CH-CS, and CH-CS-EGCG were recorded by FTIR spectroscopy (Nicolet iS5, Thermo Fisher, Waltham, MA, USA). The scanning frequency was configured to span the range of 4000 to 400 cm^−1^, while the resolution was 4 cm^−1^ [26].

### 2.7. Determination of the Antioxidant Capacity of the Films

*DPPH scavenging assay*: Film samples (0.1 g) were immersed in distilled water (10 mL) for 1 h, then centrifuged 10 min at 3000 r/min. A 3 mL aliquot of DPPH working solution (0.1 mM) was combined with 1 mL supernatant, and the mixture allowed to react in darkness for 30 min. Absorbance at 517 nm (*A*_1_) was recorded against controls containing sample in ethanol (*A*_2_) and DPPH-water mixture (*A*_0_). Antioxidant activity (%) was calculated as below [27].
(5)$$\begin{array}{c} {\mathrm{DPPH}\;\mathrm{radical}\;\mathrm{scavenging}\;\mathrm{activity}\;(\%)=(1\;-\;(A}_{1}-A_{2}{)/A}_{0})\;\times \;100\; \end{array}$$

*ABTS radical cation decolorization assay*: ABTS radical cation was generated by oxidative reaction of ABTS (7.4 mM) with potassium persulfate (2.6 mM), followed by 16 h dark incubation. The reaction mixture was standardized to an absorbance of 0.70 ± 0.01 at 734 nm. For testing, 3 mL diluted ABTS solution was mixed with 1 mL sample supernatant and allowed to equilibrate for 10 min in darkness. Measurements at 734 nm were referenced to distilled water blank. Antioxidant capacity (%) was determined using the following formula.
(6)$$\begin{array}{c} {\mathrm{ABTS}\;\mathrm{radical}\;\mathrm{scavenging}\;\mathrm{activity}\;(\%)=(1\;-\;(A}_{a}\;{-\;A}_{b}{)/A}_{c})\;\times \;100\; \end{array}$$ where *A*_a_ = absorbance of the ABTS mixture after adding the sample; *A*_b_ = absorbance of distilled water containing the sample; *A*_c_ = absorbance of the ABTS mixture with distilled water.

### 2.8. Determination of the Antibacterial Ability of Films

The *Staphylococcus aureus* colonies were activated and cultured in a constant temperature shaker at 180 rpm and 37 °C for 12 h. Taking 100 μL of the activated bacterial suspension, diluted to an appropriate concentration. The bacterial concentration was determined using the plate count method. The colony concentration was adjusted to 10^6^ CFU/mL and stored at 4 °C for later use. Using the double dilution method, the sample solutions (100 μL) were inoculated into it, respectively. Then it was incubated at 37 °C for 24 h, and then we performed plate counts. Sterile water was used as a control to calculate the antibacterial rate.
(7)$$\begin{array}{c} \mathrm{Inhibition}\;\mathrm{rate}=\frac{\left(N_{0}-N\right)}{N_{0}}\times \;100\%\; \end{array}$$ where *N*_0_ is the colonies count of the control group; *N* is the number of colonies in the treatment group.

### 2.9. Determination of the Effect of Films on Mango Preservation

Mangoes were purchased from a local supermarket, and those with uniform firmness and color were selected for the preservation experiment. After initial screening, the mangoes were divided into five groups: one control group (no coating) and four treatment groups (coated with different film solutions). For the treatment groups, mangoes were first immersed in the prepared film solution continuously, then air-dried at room temperature for 30 min until a transparent film formed on their surfaces. This coating process was repeated three times to ensure full coverage. For the control group, mangoes were placed in water for 5 min instead, followed by 30 min of air-drying, and this procedure was also repeated three times.

#### 2.9.1. Weight Loss Measurement

Weight loss during mango storage was assessed through weighing. Mango weights were measured using an electronic balance. The weight loss rate was calculated as the percentage difference between the initial and subsequent weights, relative to the initial weight. Specifically, if *m*_0_ represents the initial weight (g) and m represents weight on day n, the calculation formula is as follows:
(8)$$\begin{array}{c} \mathrm{Weight}\;\mathrm{loss}\;=\frac{m_{0}\;-m}{m_{0}}\times \;100\%\; \end{array}$$

#### 2.9.2. Firmness Measurement

The firmness of mangoes was assessed using a fruit firmness tester equipped with a 2 mm diameter cylindrical stainless steel probe. The test was conducted at a constant speed of 2 mm·s^−1^, with the probe penetrating the fruit to a depth of 2 mm. Firmness was quantified by measuring the maximum force (N) exerted during tissue rupture.

#### 2.9.3. Titratable Acidity (TA) Measurement

Mango pulp (10 g) was homogenized in 100 mL distilled water, followed by centrifugation at 5000 rpm for 10 min at 4 °C. The clarified supernatant was collected for analysis. TA was quantified by titrating the supernatant with 0.1 M NaOH (phenolphthalein indicator) until a persistent pale pink endpoint. TA (expressed as g citric acid equivalents/100 g sample) was calculated using the following formula.
(9)$$\begin{array}{c} \mathrm{TA}\left(\%\;citric\;acid\right)=\frac{V_{1}\times 0.1\times 0.064\times V_{0}}{m\times V_{2}}\times 100\; \end{array}$$ where *V*_1_ = NaOH volume consumed (mL), *V*_0_ = total extract volume (mL), *V*_2_ = titrated aliquot volume (mL), *m* = sample mass (g), and 0.064 = citric acid molar mass factor.

#### 2.9.4. Total Soluble Solids (TSS) Measurement

The TSS content of the sample was assessed using a refractometer. Before each measurement, the refractometer was wiped clean and zeroed with distilled water. The sample was then mashed, homogenized, and filtered. The supernatant was collected, and 2–3 drops were placed on the refractometer lens. The TSS values were recorded after measurement.

#### 2.9.5. Statistical Analysis

Each experiment was conducted in triplicate to ensure accuracy and reduce variability. Data analysis and visualization were performed using Origin 2021 software. Statistical significance was first determined via one-way ANOVA using SPSS 22.0. To further identify pairwise significant differences among groups, Tukey’s HSD post hoc test was applied, with *p* < 0.05 defined as the threshold for statistical significance.

## 3. Results and Discussion

### 3.1. Optimization of Preparation Conditions

To enhance the performance of the film, the influence of CH to CS mass ratios (ranging from 1:1 to 1:5) on the film’s physical characteristics was explored. Film thickness, a key factor in assessing mechanical properties, was found to be minimally affected by the CH to CS ratio, remaining between 0.076 and 0.086 mm, as shown in Figure 1A. WVP, a critical property for food preservation materials, was also examined. A lower WVP value indicates stronger water barrier properties, effectively protecting food from environmental influences [28,29]. As demonstrated in Figure 1B, the WVP of the films varied significantly depending on the CH to CS ratio. At ratios of 1:1 or 1:2, the WVP was relatively high, likely due to the hydrophilic nature of CH. When CH was present in larger quantities, it created a moist environment that caused the film structure to swell, disrupting the network and increasing WVP. Conversely, as the proportion of CS in the film matrix increased, the WVP generally decreased. This reduction was attributed to the formation of hydrogen bonds between CS and CH, which reduced the number of hydroxyl groups and decreased the film’s affinity for water vapor [19]. Comparable results were reported by Qin et al. [30], who demonstrated a reduction in WVP as the LRA concentration rose, attributed to starch-LRA hydrogen bonding interactions. Additionally, higher CS content promoted the formation of a dense network structure, enhancing the film’s water resistance [31]. Notably, films with CH to CS ratios of 1:4 or 1:5 exhibited the lowest WVP values, making them more suitable for coating applications.

Food packaging materials must exhibit adequate mechanical properties to retain their structural integrity under external forces, ensuring food protection during handling and transportation. TS and EB are key parameters used to assess the mechanical performance and flexibility of films [32]. As shown in Figure 1C, TS values of the films increased from 2.78 MPa to 8.33 MPa as the CH:CS ratio rose to 1:4. This enhancement in mechanical properties was attributed to the formation of hydrogen bonds, electrostatic interactions, and other non-covalent interactions between CH and CS. Wang et al. [33] observed a similar trend in films made from soy protein isolate and carboxymethyl konjac glucomannan (CMKGM), where TS reached its maximum when CMKGM concentration was 80 wt%, due to hydrogen bonding between the two components. When the CH:CS ratio was 1:5, TS decreased; however, this change was not statistically significant. This decline might be linked to the reduced number of carboxyl and amino groups in the composite film, which diminished the hydrogen and ionic bonding between CH and CS [34]. The EB of CH-CS composite films followed a non-linear trend as CS content increased, initially decreasing, then increasing, and finally decreasing again. At a CH:CS ratio of 1:1, the high EB value was attributed to the abundance of short peptide chains and hydrophilic groups in CH, which enhanced molecular chain mobility, imparting superior flexibility and extensibility to the film. When the CH:CS ratio reached 1:2–1:3, competitive binding between CS amino groups and CH carboxyl groups restricted CH chain mobility, reducing plasticity and leading to decreased EB. At a 1:4 ratio, the increase in CS content led to a denser film structure, contributing to improved EB. However, excessive CS (1:5) promoted preferential CS-CS hydrogen bonding over CH-CS interactions, increasing film rigidity and reducing extensibility.

The light barrier properties of biopolymer films are crucial factors affecting the quality of preserved food. Lower light transmittance indicates a stronger light barrier capability, effectively reducing the impact of photo-oxidation on food quality [35]. Figure 1D showed that when the CH:CS mass ratio was 1:1, the light transmittance was as high as 43.47%, indicating good compatibility between the membrane matrix. The light transmittance of the composite films exhibited a gradual decrease with decreasing CH:CS mass ratio, reaching minimal values at ratios of 1:4 and 1:5. This might be attributed to an excess of chitosan, leading to molecular chain rearrangement and thus enhanced light absorption and scattering in the film. Considering all the data, the composite film with a CH:CS mass ratio of 1:4 exhibited both low water vapor permeability and low light transmittance, while also possessing sufficient mechanical strength. Therefore, this CH:CS mass ratio of 1:4 was selected as the optimal preparation condition for the edible coating preservation film in this study.

### 3.2. Structural Characterization of the Films

The microstructure of the films offers insights into their composition distribution, which is critical for understanding their physical properties [36]. As depicted in Figure 2, the CH-CS films exhibited a smooth and flat surface, reflecting good compatibility between CS and CH. However, the presence of cracks indicated limitations in their mechanical performance. The incorporation of EGCG improved the film’s properties. At a low EGCG content (1 wt%), it was evenly distributed within the matrix, resulting in reduced cracking. When EGCG was increased to 2 wt% or 3 wt%, the cracks disappeared, but the surface became rougher and less uniform. This change might be attributed to the interaction between EGCG and the film matrix, where excessive aggregation of components reduced film fluidity and increased rigidity. Riaz et al. [37] documented surface irregularities in chitosan-based matrices incorporating apple peel polyphenolic extracts. The presence of textural heterogeneities may functionally enhance interfacial stabilization within food systems, thereby augmenting structural resilience against physical stressors [38]. Thus, the addition of EGCG increased surface roughness, suggesting its potential for use in food preservation packaging.

The structural changes and molecular interactions in the composite films were characterized using FT-IR spectroscopy, as shown in Figure 3. The incorporation of EGCG into the CH-CS matrix did not introduce new absorption peaks but caused a shift in the peak positions, which suggests successful embedding of EGCG within the CH-CS composite film. The amide A band of CH-CS was observed at 3243 cm^−1^. After incorporating 1 wt%, 2 wt%, and 3 wt% EGCG, this band shifted to 3240 cm^−1^, 3236 cm^−1^, and 3233 cm^−1^, respectively. The gradual decrease in wavenumber (up to 10 cm^−1^) confirms enhanced hydrogen bonding [39,40]. This bonding occurs between EGCG’s phenolic hydroxyl groups and the N-H/O-H groups of CH-CS. The Amide B band (assigned to C-H stretching vibrations) was detected at 2924 cm^−1^ in CH-CS. Upon EGCG addition, it shifted slightly to the range of 2923–2925 cm^−1^. This minor shift reflects hydrophobic associations [41]. Specifically, it arises from interactions between EGCG’s benzene rings and the hydrophobic moieties (e.g., acetyl groups) of CH-CS. In CH-CS, the Amide I band (related to C=O stretching vibrations) and Amide II band (related to N-H bending vibrations) were located at 1635 cm^−1^ and 1537 cm^−1^, respectively. With the addition of EGCG, both bands shifted: the Amide I band moved to 1634 cm^−1^, and the Amide II band shifted to 1538–1540 cm^−1^. This further verifies electrostatic interactions, which occur between the positively charged amino groups of CS and the negatively charged groups of CH [42]. A similar interaction mechanism was reported by Xie et al. [43], who noted that EGCG was encapsulated in Zein-Lecithin nanocomplexes through a combination of hydrophobic, electrostatic, and hydrogen bonding interactions. Overall, the interaction between EGCG and CH-CS is primarily mediated by hydrogen bonding, with additional contributions from electrostatic forces, and hydrophobic interactions.

### 3.3. Physical Properties of the Films

As shown in Figure 4A, although different EGCG concentrations significantly affected the thickness of CH-CS films, the thickness of the films only slightly increased from 0.094 to 0.111 mm. This slight change can be attributed to EGCG replacing water in the films, thereby increasing the proportion of dry matter [22,23]. The WVP values of the composite films are presented in Figure 4B. The addition of EGCG into the CH-CS matrix resulted in a significant reduction in WVP. FT-IR analysis revealed the presence of intermolecular hydrogen bonding between EGCG and the film matrix, which reduced the number of hydrophilic groups available for interaction, thereby lowering the film’s WVP [35,44]. Furthermore, the interaction between EGCG and CH-CS enhanced the internal binding forces, leading to a more compact network structure that restricted water vapor transmission [31]. A similar observation was made by Yadav et al. [45], who reported that chitosan molecules could form a condensed macromolecular network with ZnO@gal, resulting in reduced WVP. Films with lower WVP values can effectively reduce water exchange between food and its surroundings, thereby extending shelf life. For instance, a study focused on EGCG-loaded chitosan/pectin active films demonstrated that EGCG enhanced the matrix’s resistance to water molecule transport. This enhanced resistance led to a reduction in WVP. Ultimately, it extended the shelf life of strawberries by 4 days [35]. Ultraviolet (UV) light is a significant factor causing oxidative damage to food products. Given the role of these films as packaging materials for mango preservation, it is essential to assess their capacity to block UV radiation. As shown in Figure 4C, the films incorporated with EGCG exhibited comparable UV transmittance (<35%), indicating a certain level of UV-blocking capacity. The lack of a concentration-dependent effect suggested that the EGCG loading levels (1–3 wt%) were below the threshold required for further optical modification.

### 3.4. Antioxidant and Antibacterial Activities of the Films

Antioxidant activity plays a critical role in assessing the effectiveness of food preservation, with enhanced antioxidative properties being essential for extending shelf life [46]. The radical scavenging ability of the edible coating was assessed through DPPH and ABTS assays. As illustrated in Figure 5A, the CH-CS coating demonstrated modest antioxidant performance, achieving DPPH and ABTS radical scavenging rates of 12.40% and 26.82%, respectively. This activity was primarily attributed to the bioactive components of CS and CH. The antioxidative properties of CS films were linked to the presence of free amino and hydroxyl groups in the CS structure, which facilitate radical quenching [47]. He et al. [48] reported that hydrolyzed collagen releases bioactive peptides that effectively inhibit lipid oxidation and neutralize free radicals. Interestingly, the radical scavenging capacity of the edible coating significantly improved with increasing EGCG concentration. The ABTS and DPPH scavenging efficiencies of the film loaded with 3 wt% EGCG were 234% and 422% higher, respectively, compared to the CH-CS film. These findings underscore the exceptional antioxidant performance of the EGCG-incorporated edible coating, highlighting its potential as an effective food preservation solution.

Food preservation materials with antimicrobial components can effectively prevent microbial invasion and disrupt microbial structures by releasing active substances, thereby prolonging the shelf life of food [49]. In this study, the antibacterial properties of the edible coating film were evaluated using *S. aureus* as a representative model. As shown in Figure 5B, the CH exhibited limited antibacterial activity, which was attributed to the inherent antimicrobial properties. The antibacterial efficiency of the CH-CS-EGCG film was found to increase with the EGCG content. When 2 wt% or 3 wt% EGCG was added, the antibacterial performance of the composite film was significantly improved. Previous research by Cui et al. [50] demonstrated that films enriched with EGCG displayed inhibitory effects on Gram-positive bacteria, with the inhibitory effect being closely associated with the EGCG concentration. Taking all factors into consideration, a composite film loaded with 3 wt% EGCG was chosen for subsequent mango preservation experiments.

### 3.5. Formation Mechanism of the Films

EGCG is a polyphenolic compound that possesses multiple phenolic hydroxyl groups in its molecular structure. These hydroxyl groups can engage in interactions with other functional groups within the film matrix, forming physical networking. CH-CS-EGCG films with varying EGCG content were fabricated using the flow film method (Figure 6). During the formation of the films, the interactions between EGCG and the film matrix play a key role in forming the film [35]. The phenolic hydroxyl groups of EGCG engage in hydrogen bonding with the hydroxyl, carboxyl, and amino functional groups in CS and CH, improving intermolecular affinity. The benzene rings in EGCG interact with the hydrophobic regions in the matrix (such as the acetyl groups of CS or the hydrophobic amino acid side chains in CH) via hydrophobic associations. Additionally, the positively charged CS draws the negatively charged CH via electrostatic interactions. These interactions not only enhance the cross-linking density of the film but also substantially improve the mechanical properties and water resistance of the film. During the flow film preparation process, EGCG molecules are uniformly dispersed in the solution with the matrix molecules. As the solvent evaporates, intermolecular interactions gradually intensify, leading to the self-organization of molecules into an ordered network structure [12]. Furthermore, as a functional substance, the incorporation of EGCG imparts various biological activities, such as antioxidant and antibacterial properties, thereby broadening the application scope of the film.

### 3.6. Preservation of Mangoes

#### 3.6.1. Appearance of Mangoes During Storage

The CH-CS film loaded with 3 wt% EGCG was used for mango preservation to explore the application of the edible coatings. During storage, the most obvious visual changes in mangoes are softening of the peel, wrinkling, and the appearance and expansion of black spots on the skin [51]. To visually evaluate the effect of CH-CS-EGCG films on mango pretreatment, mangoes coated with different film solutions (CH, CS, CH-CS, CH-CS-EGCG) were tested. For complete coverage of the film solutions on the surface of the tested mangoes, dip-coating was performed three times. This dip-coating process is easy to operate and suitable for the laboratory stage; for industrial production, spray coating can be adopted as an alternative. The visual changes in mangoes during storage are shown in Figure 7. The uncoated group began to turn dark yellow and decay after 4 days, with wrinkled skin and black spots. By day 10, the mangoes were completely rotten and visibly moldy. CH coating showed almost no preservation effect on mangoes. Compared with the control group of mangoes, there was little difference in appearance during storage. This could be attributed to the inadequate film-forming ability of CH, which hindered the creation of an effective thin film barrier on the mango surface. Mangoes treated with other composite film solutions showed slower visual changes, with the composite film containing 3 wt% EGCG exhibiting the best preservation effect. Until the 10th day, the mangoes coated with CH-CS-EGCG_3%_ still maintained their yellowish-green color and plump appearance, effectively extending the shelf life by 6 days. This result was far superior to the 5-day shelf life of mangoes packaged with PE film [52]. The CH-CS-EGCG edible coating film served as a protective barrier for mangoes, effectively inhibiting microbial growth and enhancing preservation effectiveness.

#### 3.6.2. Weight Loss

Fruit weight loss during storage is a natural phenomenon primarily caused by water evaporation due to respiration and transpiration, which leads to fruit shrinkage and spoilage [53]. As shown in Figure 8A, mangoes without any treatment were directly exposed to the external environment, resulting in rapid dehydration, visible wrinkling, and tissue collapse. Mangoes coated with CH alone exhibited minimal differences compared to the untreated group, while the CH-CS coating demonstrated improved performance. This enhancement was attributed to the formation of cross-links between CH and CS, which reduced film permeability and minimized the exchange of substances between mangoes and their surroundings, thereby slowing down the rate of water loss [54]. Although the weight loss rates increased over time across all groups, mangoes coated with CH-CS-EGCG_3%_ exhibited the lowest weight loss, with only a 12% reduction. This improvement was due to the enhanced water resistance of the composite film. Its lower WVP reduced surface water loss and extended storage duration. Previous studies have shown that films with lower gas permeability and WVP are more effective in prolonging fruit shelf life [55].

#### 3.6.3. Firmness

The firmness of mango is a key indicator of its intrinsic quality, with higher firmness indicating greater integrity of the cell wall, which plays a crucial role in defense against pathogens during storage [56]. Experimental results, as shown in Figure 8B, indicate that the firmness of all mango groups decreased over time during storage. The untreated group and the group coated with CH alone experienced the most significant firmness reduction, dropping from an initial 5.59 N to 0.61 N and 1.02 N, respectively. This phenomenon was attributed to water loss in mango tissues and microbial activity on the fruit’s surface, which compromised its structural integrity [57]. It is noteworthy that the firmness of mangoes coated with CH-CS-EGCG film also decreased but remained at 3.2 N by the 10th day. This was due to the enhanced physical barrier performance of the film, which reduced mango respiration, inhibited hydrolytic enzyme activity, and slowed down the softening process, thereby preserving the “firm yet juicy” texture unique to ripe mangoes [58]. Additionally, Carvalho et al. [59] found that a CH coating enriched with cinnamaldehyde reduced hydrolytic enzyme activity in fresh-cut melons and maintained their firmness. These findings suggest that edible coatings can effectively preserve the hardness of harvested mangoes.

#### 3.6.4. Titratable Acidity

Titratable acidity (TA), an important indicator of fruit quality, also decreased in all mango groups during storage, as shown in Figure 8C [31,57]. The untreated group exhibited the most pronounced decline in TA, dropping from 0.230% to 0.052% by the 10th day of storage. In contrast, mangoes coated with CH-CS-EGCG retained significantly higher TA levels, ending at 0.127%. This difference was attributed to the ability of the CH-CS-EGCG film to form a physical barrier on the mango surface, thereby reducing respiration rates and inhibiting microbial activity [60]. These effects minimized TA consumption and helped maintain fruit quality [61]. The superior preservation capabilities of CH-CS-EGCG films, as evidenced by their ability to extend mango shelf life, were further confirmed by these findings.

#### 3.6.5. Total Soluble Solids

Total soluble solids (TSS) indicate fruit ripeness and taste. TSS levels represent the sum of sugars and organic acids in the fruit, significantly impacting its palatability. As depicted in Figure 8D, TSS levels consistently rose across all groups. During mango storage, the respiration rate steadily increased, accompanied by continuous water loss, factors that collectively contributed to the rise in TSS levels [62]. The uncoated mangoes displayed the most significant TSS variability, whereas the CH-CS-EGCG-coated group maintained a more stable TSS level. This stability was likely due to the outstanding antimicrobial and barrier characteristics of the CH-CS-EGCG coating. Research has shown that edible coatings can mitigate drastic fluctuations in TSS levels, maintaining them at lower levels to delay fruit ripening by reducing respiration rates [63]. Wu et al. [64] demonstrated that EGCG can directly influence the respiratory pathway of apples, thereby extending their shelf life. Dong and Wang [54] investigated the use of a guar gum (GG) and ginseng extract (GSE) blend as an edible coating for sweet cherries. Their findings revealed that the GG-GSE coating minimized weight loss, delayed respiratory rate changes, and slowed TSS progression.

## 4. Conclusions

In this study, an edible coating film CH-CS-EGCG was developed and applied for mango preservation. The optimal formulation was achieved by optimizing the mass ratio of CH-CS (1:4) and incorporating EGCG at 3 wt%. FT-IR analysis revealed that the interactions between EGCG and CH-CS were facilitated by hydrogen bonding, hydrophobic effects, and electrostatic forces. Functionally, the film exhibited significant antioxidant and antibacterial properties, which were primarily attributed to the concentration of the EGCG in the matrix. When applied to mangoes, the film containing 3 wt% EGCG demonstrated exceptional protective capabilities, effectively extending the shelf life of the fruit by 6 days. These findings underscore the potential of CH-CS-based films loaded with EGCG as an innovative edible coating solution, offering a sustainable approach to enhance the shelf life of perishable fruits and contribute to eco-friendly food packaging systems. However, the relatively high price of raw materials may limit its competitiveness compared with traditional plastic packaging. Further work will focus on exploring low-cost alternative sources of collagen hydrolysate to enhance its practical applicability.

## Figures and Tables

**Figure 1 foods-14-03609-f001:**
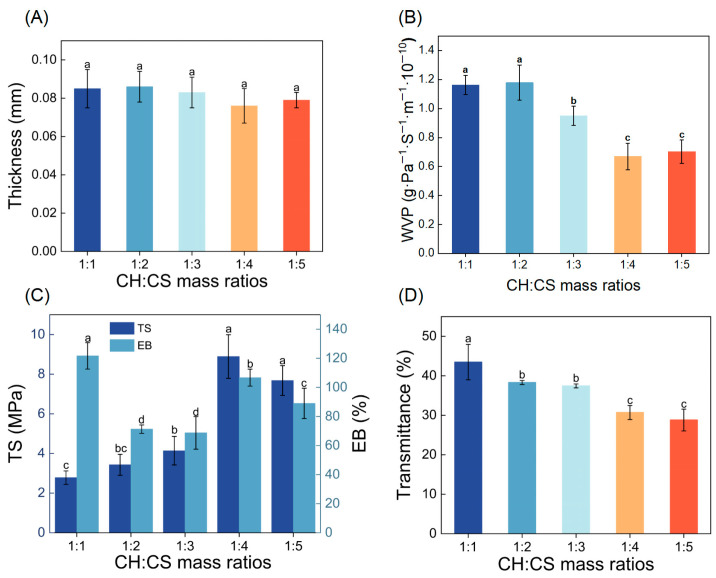
The physical properties of CH-CS films. Thickness (**A**), WVP (**B**), TS and EB (**C**), Light transmittance (**D**). Different lowercase letters over the bar indicate significant differences (*p* < 0.05).

**Figure 2 foods-14-03609-f002:**
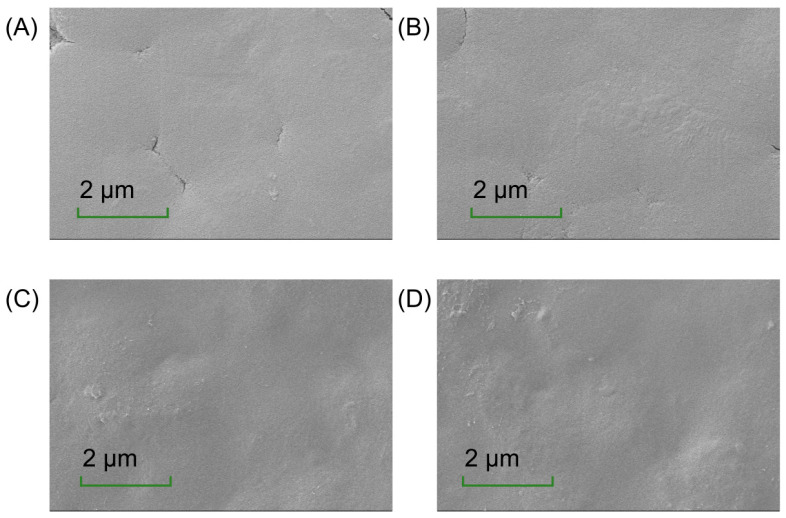
The surface morphology of CH-CS (**A**), CH-CS-EGCG_1%_ (**B**), CH-CS-EGCG_2%_ (**C**), CH-CS-EGCG_3%_ (**D**). Images acquired at a magnification of 5000×. Scale bar: 2 µm.

**Figure 3 foods-14-03609-f003:**
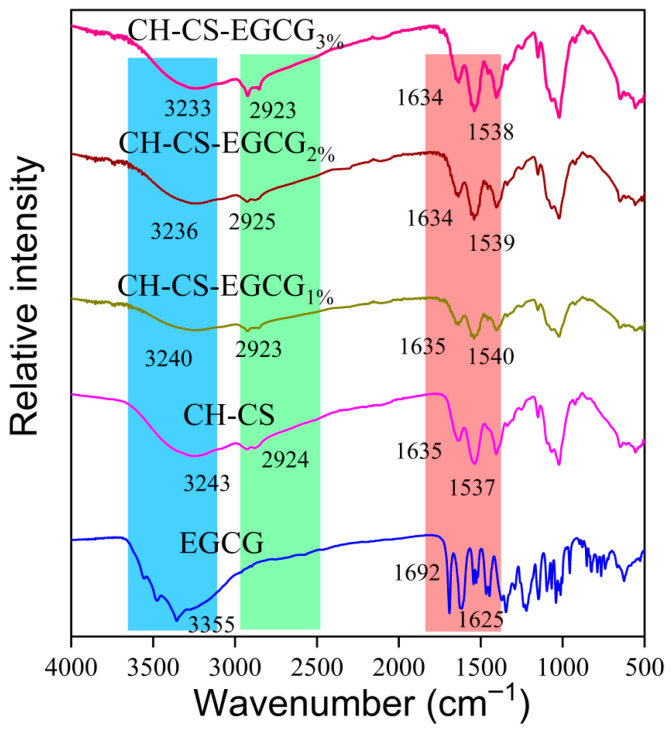
The FTIR spectra of EGCG, CH-CS, and CH-CS-EGCG.

**Figure 4 foods-14-03609-f004:**
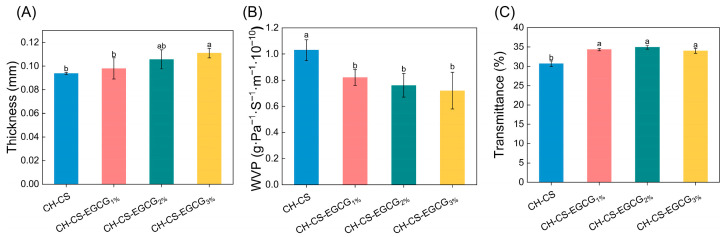
The physical characterization of CH-CS and CH-CS-EGCG films. Thickness (**A**), WVP (**B**), Light transmittance (**C**). Different lowercase letters over the bar indicate significant differences (*p* < 0.05).

**Figure 5 foods-14-03609-f005:**
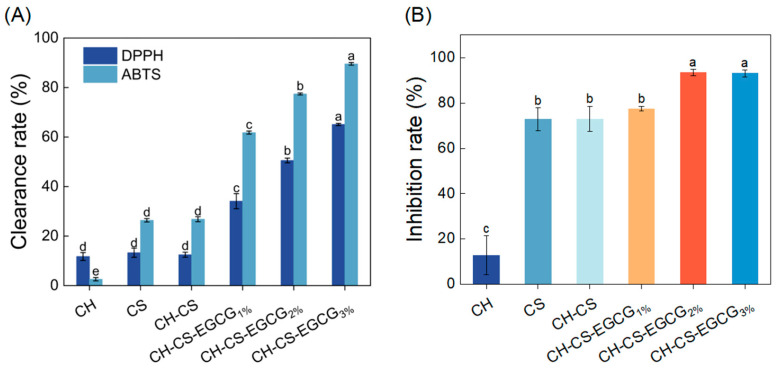
The DPPH and ABTS free radical scavenging activity of CH, CS, CH-CS and CH-CS-EGCG (**A**). The inhibition rate of CH, CS, CH-CS and CH-CS-EGCG to *S. aureus* (**B**). Different lowercase letters over the bar indicate significant differences (*p* < 0.05).

**Figure 6 foods-14-03609-f006:**
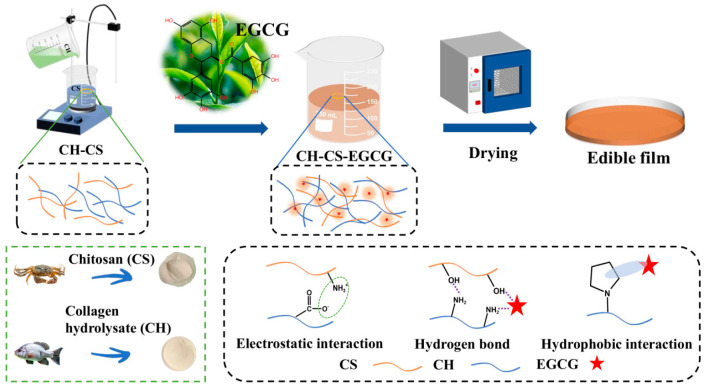
The preparation and formation mechanism of the CH-CS-EGCG film.

**Figure 7 foods-14-03609-f007:**
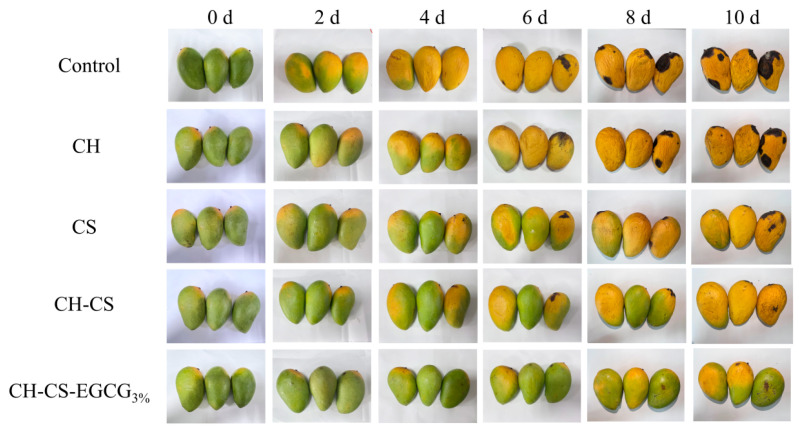
The photographs of mangoes coated with edible films during storage at room temperature.

**Figure 8 foods-14-03609-f008:**
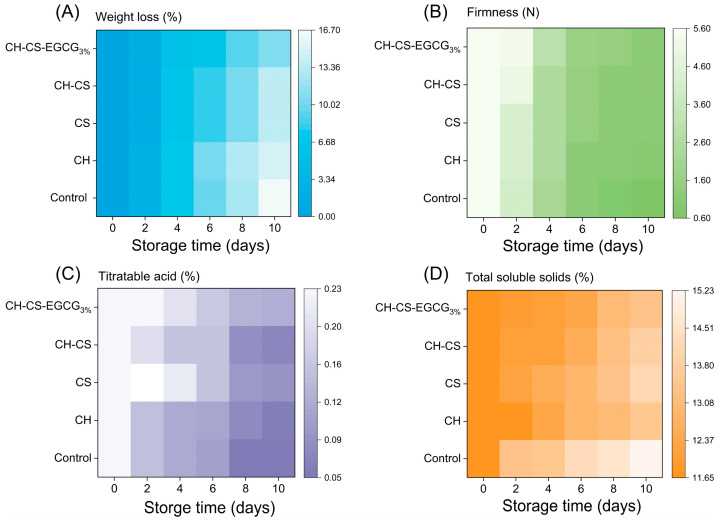
The mangoes preservation indexes of CH-CS-EGCG film during 10 days of storage at room temperature. Weight loss (**A**), Firmness (**B**), Titratable acid (**C**), Total soluble solids (**D**).

## Data Availability

The original contributions presented in this study are included in the article. Further inquiries can be directed to the corresponding authors.

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
