# Peer review of "Nature’s Preservative: Epigallocatechin Gallate-Loaded Edible Film Extends Mango Shelf Life"

_foods, 2025, doi:10.3390/foods14213609_

Round 1

Reviewer 1 Report

Comments and Suggestions for Authors

Recommendations for Authors

Introduction

  • Background is thorough and references are current.
  • Add more context on global mango post-harvest losses outside China to strengthen relevance.

Research Design

  • Design is appropriate. Control and multiple coating formulations are well chosen.

Methods

  • Methods are clearly described and replicable.
  • Consider clarifying the statistical analysis section: specify exact post-hoc tests after ANOVA.

Results

  • Results are clearly presented with adequate figures and tables.
  • Highlight any limitations, such as scalability of film preparation or potential cost issues.

Conclusions

  • Supported by the data.
  • Could benefit from a brief discussion on commercial application challenges.

Figures/Tables

  • All figures and tables are clear and labeled.

Quality of English

English is fine and needs only minor polishing.

  • The edible CH-CS-EGCG film shows strong antioxidant (up to 422 % higher DPPH scavenging) and antibacterial activity, with clear impact on mango shelf-life extension (six days) .
  • Discuss potential sensory effects of the coating on fruit taste and texture.
  • Provide cost analysis or scalability notes to guide practical adoption.
  • Include shelf-life comparison with existing commercial coatings for context. 
Comments on the Quality of English Language

The manuscript is well written in clear, fluent English. Only minor copy-editing for grammar and style is suggested.

Reviewer 2 Report

Comments and Suggestions for Authors

Comments

  1. In The section is generally well-written, but certain sentences are dense. For example, lines 46–48 (“via hydrogen bonding, hydrophobic, and electrostatic interactions…”) could be simplified to improve readability. Consider breaking long sentences into two for better flow.
  2. The description of EGCG’s role (“crosslinks chitosan via non-covalent interactions”) could be expanded with more detail. Is this mainly hydrogen bonding or hydrophobic interaction? Adding mechanistic insight would strengthen the scientific rigor.
  3. The study uses one-way ANOVA (p < 0.05), but post-hoc tests (e.g., Tukey’s) are not mentioned. Without these, it is unclear which groups differ significantly.
  4. The FT-IR results indicate hydrogen bonding and hydrophobic interactions, but no quantitative analysis (e.g., peak shift values) is provided. Including detailed spectral assignments would strengthen the mechanistic interpretation.
  5. The proposed "formation mechanism" (Figure 6) is largely descriptive. Could the authors provide schematic molecular interactions or refer to molecular dynamics simulations reported in similar studies?
  6. Figures: Several figures (SEM, FT-IR, preservation results) are referenced, but resolution and clarity should be improved for readability.]
  7. Abstract: Avoid giving too many numerical values; focus on key findings.

Reviewer 3 Report

Comments and Suggestions for Authors

Numerous studies have been published in the literature regarding the preparation, characterization, and application of antioxidant packaging films based on chitosan-epicatechin gallate conjugates. In this study, an additional CH-CS-EGCG coating was prepared and characterized, and its ability to slow the decay and spoilage of mango, a highly susceptible fruit, was evaluated, extending its shelf life by 6 days.

This research provides a valuable reference for the study of edible coatings for the preservation of this fruit as an alternative to traditional preservation methods, namely freezing and chemical treatments. Perhaps a comparison with freezing would have been appropriate, as well as an evaluation of packaging for protecting the fruit during transport.

Round 2

Reviewer 2 Report

Comments and Suggestions for Authors

Author has implimentated the comments which raise in revised version of manuscript and i please with this form of revision,  accept in as such form of manuscript 

Author Response

Comments 1: Author has implimentated the comments which raise in revised version of manuscript and i please with this form of revision, accept in as such form of manuscript.

Response 1: Thank you very much for your positive feedback on it.